# Psychosocial and Health-Related Behavioral Outcomes of a Work Readiness HIV Peer Worker Training Program

**DOI:** 10.3390/ijerph20054322

**Published:** 2023-02-28

**Authors:** Erin McKinney-Prupis, Yung-Chen Jen Chiu, Christian Grov, Emma K. Tsui, Sharen I. Duke

**Affiliations:** 1Alliance for Positive Change, New York, NY 10001, USA; 2Hunter College, City University of New York, New York, NY 10065, USA; 3Graduate School of Public Health and Health Policy, City University of New York, New York, NY 10017, USA

**Keywords:** HIV, peer, work, employment, health

## Abstract

Targeted work readiness training is an important approach to help people living with HIV (PLHIV) to overcome their unique barriers to work, while addressing social determinants of health needs. This study assesses the psychosocial impacts of a work readiness training and internship program among HIV peer workers in New York City. From 2014 through 2018, 137 PLHIV completed the training program, and 55 individuals completed both the training and the six-month peer internship. Depression, HIV internalized stigma, self-esteem, HIV medication adherence, patient self-advocacy, and safer sex communication apprehension were used as outcome measures. Paired *t*-tests were performed to determine if significant score changes occurred at the individual level before and after each training. Our results show that participating in the peer worker training program significantly decreased depression and internalized HIV stigma, and significantly increased self-esteem, medication adherence, and patient self-advocacy. The study underscores that peer worker training programs are important tools to improve not only the work readiness of PLHIV, but also psychosocial and health outcomes. Implications for HIV service providers and stakeholders are discussed.

## 1. Introduction

Due to medical advancements, an HIV diagnosis has changed from a terminal illness to a chronic illness and emergent disability [1,2]. Despite medical conditions, people living with HIV (PLHIV) often experience various social and economic challenges, such as poverty, discrimination, unstable housing, and unemployment/underemployment. These social, economic, and political systems that can impact individual health risks and outcomes are defined as social determinants of health (SDH) [3]. SDH can also be referred to as upstream determinants of health, or can be more proximal indicators such as geographic areas, socioeconomic status, education, access to healthcare, and employment status. A SDH framework has been adopted by the United States Centers for Disease Control and Prevention in addressing health-related disparities [4]. PLHIV have fewer health-related barriers to work than ever before, yet the employment rate among PLHIV in 2019 was estimated at 48.6%, compared to 63.1% in the general population [5,6]. Peer intervention (or peer education) has been identified as an effective approach in various fields, such as mental health, substance abuse, and HIV care [7,8]. Peer workers play an important role in HIV prevention and treatment. Participating in HIV peer worker training not only facilitates vocational development, but also enhances their psychosocial and health-related behaviors. Nevertheless, outcomes related to HIV peer worker training have not yet been studied [8,9].

### 1.1. HIV Peer Intervention and Peer Worker Training

Peer workers (or peer providers, peer specialists, peer educators) are those who have lived experiences and who work in the healthcare or social service industries [8,10]. Peer workers share similar demographic characteristics (gender, age, race/ethnicity), lived experiences (i.e., men who have sex with men, injection drug use, commercial sex work), health conditions (HIV, diabetes, obesity, mental illness, alcohol and other substance use) and/or community memberships with their clients [8,10]. Peer workers can access communities in different ways than medical and behavioral health professionals; sometimes they are the only people with an ability to gain entry to a community, such as sex workers. Peer workers influence individuals in a unique way, as they have experienced similar challenges and life experiences as a client/patient themselves [11]. Peer interventions have been incorporated into HIV treatment and prevention programs since the inception of the HIV epidemic [9]. HIV peer workers provide a variety of services including health education, psychosocial support, harm reduction, community outreach, navigation to medical appointments, and treatment adherence [12]. Simoni et al. conducted a systemic review of 117 studies of the efficacy of HIV peer intervention and identified positive outcomes, including increased HIV knowledge, decreased sex risk behaviors, and decreased substance use [9]. Other systemic review studies also indicated that peer intervention lead to decreased HIV-related stigma, increased HIV testing rates, medication adherence, retention in care, self-efficacy, and quality of life [13,14].

Over the past two decades, peer interventions have become more structured and the roles of peer workers have become more formalized. State and national certifications have been created. Starting in 2007, states with peer certification processes could bill Medicaid for peer delivered behavioral health services and, in 2014, states could bill for health and wellness services [15]. As a result of the 2015 New York State’s Blueprint for Ending the Epidemic by 2020, a New York State (NYS) peer worker certification was created [16,17]. The purpose of the certification is to professionalize HIV peer work, to increase work opportunities for PLHIV, and to expand the utilization of a professional peer workforce to help achieve the state’s goals of improving linkage and retention in care, increasing rates of viral suppression, and preventing new infections [16]. The process of becoming a peer worker also facilitates work-related skills development and provides work experiences for PLHIV. Research identified that the training of peer workers enables them to obtain competitive employment in the community. Effective peer worker training also enhances program sustainability, as it ensures quality services provided by peer workers [18]. There has been research on the impacts of peer intervention on PLHIV, but limited research focusing on HIV peer workers.

### 1.2. Employment as a Social Determinant of Health

Employment has been recognized as an important social determinant of health, which has significant associations on the health and wellness of PLHIV. Employment impacts a variety of factors related to financial and life stability, including earned income, access to healthcare, access to housing, and social connectedness; these factors directly or indirectly influence individuals’ health, wellbeing, and quality of life. Maulsby et al. conducted a scoping review study on employment and HIV and found that employment status is associated with HIV continuum of care outcomes, including HIV testing, linkage-to-care, retention-in-care, and medication adherence [19]. Studies have found that employed PLHIV have a higher quality of life than those who are unemployed, even after controlling for CD4 and viral load [20,21,22]. Employed PLHIV have significantly better health and prevention outcomes, such as increased CD4 counts, perceived health and psychological well-being, memory skills, and medication adherence [23,24,25,26,27,28].

Meanwhile, individuals who have undesirable employment outcomes (job loss and underemployment) often experience negative prevention and health outcomes, such as an increased risk for drug and alcohol use, and developing psychological problems. The quality of employment (job security, salaries, and job demands, control, and support) also has an impact on the physical and mental well-being of PLHIV [29]. For example, Reuda et al. found that those who had higher levels of job security reported better mental health outcomes among men living with HIV [30]. Nevertheless, individuals who had high psychological demand and unstable jobs experienced similar depression and quality of life outcomes as the unemployed individuals. Research also identified some predictors of desirable employment outcomes among PLHIV, including demographic characteristics, education, health status, beliefs, and psychosocial factors [19]. Studies show that PLHIV who have poorer HIV health outcomes (CD4 count, viral load, HCV co-infection) and lower mental health functioning are less likely to be employed [21,31]. However, HIV health status is not the only barrier preventing PLHIV from returning to work; other barriers to employment include loss of benefits, reduced housing subsidies, adequate health insurance, stigma, substance use, out-of-date job skills, and large gaps in employment history [32,33,34].

Goldblum and Kohlenberg developed the client-focused considering work model, which is the first model to conceptualize factors and processes associated with considering the work of PLHIV [35]. Conyers revised the model to apply to all individuals with emerging or episodic illnesses [36]. This model integrates the transtheoretical model of change (stages of change model) to address different phases of considering work status: contemplation, preparation, action, or resolution [37]. According to the model, individuals go through these four phases of considering work [35]. The model emphasizes the importance of weighing the impacts of returning to work on the medical, psychosocial, financial/legal, and vocational domains of influence as individuals work through the phases [35]. The model stresses that the decision to work is made by the client; choosing not to work does not suggest failure. For some PLHIV, employment may not be the appropriate outcome and therefore, capturing the impact on employment mediators is critical to assessing program effectiveness. To increase federal and state funding for vocational interventions for PLHIV, and to decrease the large unemployment rates amongst PLHIV, HIV service organizations need to evaluate and publish their vocational development interventions, including peer worker training and peer internship programs.

### 1.3. Vocational Development and Intervention for People Living with HIV

Vocational development and intervention programs are aimed at helping people re-enter or enter the workforce through structurally based techniques. Conyers and Boomer examined the relationship between the receipt of vocational rehabilitation services and health and prevention outcomes for PLHIV [38]. The study found that receiving vocational rehabilitation services was associated with better health outcomes and lower risky transmission behaviors. Unfortunately, research has also found that PLHIV severely underutilize services from state VR systems [39,40,41]. In 2009, Jung and Bellini found that PLHIV were four to six times less likely to access VR services [40]. One of the possible reasons of the underutilization of VR services is because PLHIV feel more comfortable receiving services at HIV service organizations [42]. HIV-related stigma and use of disability benefits also influence use of VR service among PLHIV. In a mixed-method study, participants not receiving social security benefits (SSI) or social security disability benefits (SSDI) discussed how the anticipated stigma of being a benefits recipient would prevent them from entering the workforce. Nevertheless, those who have received SSI or SSDI experienced the perceived stigma of receiving benefits and expressed anxiety related to losing their benefits when returning to work [43].

Since the introduction of highly active antiretroviral therapy (HAART), a variety of evaluation articles on US-based vocational development programs for PLHIV have been published [42,43,44,45,46,47,48,49,50,51]. The interventions were based on various theoretical frameworks and aim to help PLHIV obtain part-time or full-time employment, as well as assist them with return-to-work efforts such as volunteering, job training, and education. These vocational development programs provided a range of services, including vocational assessments, assistance with the development of employment goals, assistance with job search, resume writing, and interview techniques [42,43,44,45,46,47,48,49,50,51]. HIV peer worker training applies the structured work readiness approach which aims at preparing individuals for the unique challenges of work associated with living with HIV. Work readiness programs often address HIV and disability legal rights, benefits counseling, assistance towards appropriate disclosure, stigma, managing medications and appointments while working, safer sex, nutrition, and patient-physician communication [45,46,47,48,49]. In addition, some programs discussed managing mental health and disabilities and some included sessions aimed at soft skills, such as time management and communication skills [45,47,48,49]. Many vocational development programs utilized supported employment, also known as the place-and-train model [52,53]. These programs aimed to place their participants into competitive employment as soon as possible and provide on-the-job support [50,54,55,56]. Targeted work readiness programs are an important approach to help PLHIV overcome their unique barriers to work, such as the psychosocial impact of years of unemployment/no work experience, adjusting to independence from entitlements, navigating disclosure, confidentiality, properly using leave time, and negotiating reasonable accommodations [52,53]. HIV peer worker training is a unique vocational training model that helps PLHIV gain knowledge and experience to become peer workers, while addressing their unique challenges. To date, there is limited research on this model.

In program development and evaluation, mediators are the variables a program aims to directly impact, which in turn influence the program’s overall projected outcomes [57,58,59]. In the example of vocational development programs for PLHIV, mediators are the various psychological (i.e., self-esteem, self-efficacy, stigma) and job readiness skills (i.e., job seeking skills, soft skills, problem-solving ability) that are assumed to impact whether someone can search, obtain, and sustain employment. The lack of assessment of important employment mediators in vocational development interventions is a large gap in the HIV employment literature. These mediators were components of many of the above programs but were only assessed in three of the interventions [46,47,49]. Vocational development programs need to be examined with a wider lens, enabling a better understanding of how they are shaping PLHIV, as well as if they are helping PLHIV get back to work.

To this end, our research question was: were there any significant changes in participants’ psychosocial outcomes (depression, internalized HIV stigma, self-esteem) and health-related behavioral outcomes (medication adherence, patient self-advocacy, safer sex communication apprehension) between baseline, targeted work readiness peer worker training program completion, and peer internship completion. The authors aimed to understand not only the impact of the peer worker training program, but also the additive effects of participating in both the peer worker training program and the peer internship.

## 2. Methods

This study involved a retrospective secondary data analysis of an evaluation conducted between 2014 and 2018 of a work readiness HIV peer worker training and peer internship program. The purpose of the study was to examine the psychosocial and health related behavioral outcomes among 259 individuals living with HIV that participated in the peer worker training program. This was the first longitudinal evaluation efforts of this peer worker training program.

### 2.1. Peer Worker Training Program

This New York City-based peer worker was a comprehensive eight-week training that took place three days a week from 10 am to 4 pm. Training. The training evaluated in this paper was developed by Alliance for Positive Change in the mid-1990s as a foundational peer worker training and later used to help prepare individuals to take state recognized certificate peer trainings, such as NYS HIV peer certification training. The training was called: Peer Recovery Education Program, PREP, and will be entering its 60th cycle of training in 2023. To qualify for the training, individuals must have been diagnosed with HIV, at-risk for HIV, or had close relationships with PLHIV. HIV-positive participants were required to adhere to their HIV medications, compliant with primary care physician appointments, stably housed, and in recovery from drugs and alcohol for at least nine months. The training program focused on work readiness skills necessary to become health educators and patient navigators. Participants learned HIV service skills such as prevention education, presentation skills, treatment techniques, medical advocacy, and outreach and counseling techniques. Participants received training in HIV-related knowledge and its common co-infections, Hepatitis C, STIs, safer sex, substance use, and harm reduction. Participants also learned self-management skills related to, and unrelated to their HIV, such as medication adherence, safer sex skills, medical advocacy, communication skills, and stress management. The following soft skills were learned and applied: getting to work on time, following a set schedule, speaking in a group, getting along with others, and dressing appropriately. Each training cohort involved 25–40 trainees. Lastly, individuals were mandated to participate in weekly support groups.

### 2.2. Peer Internship Program

After completing the peer worker training, participants were eligible to apply to enter the peer internship program. The peer internship program functioned in six-month cycles, starting each January and July, and employed approximately 125–150 peer interns at any given time. Interns received an internship placement within the agency and participated in mentor guidance and weekly peer support groups. Participants gained work experience and applied skills learned in their foundational training course. Peers were involved in various direct and indirect HIV service positions, such as medical navigation, food and nutrition services, counseling and testing services, and education and support group facilitation. Depending on the peer placement and the type of public benefits received, a peer worked 10–20 hours a week and earned $15 per hour, up to $1300 each month. Approximately ten agency-trained participants and five other agency-trained participants were hired each cycle. New hires were based on current program needs and available positions. This work met the internship requirements for the New York State HIV peer worker certification.

The training curriculum provided participants with self-management skills and experiences aimed at impacting specific types of self-esteem. Morris Rosenberg, the author of the most widely used scale to assess self-esteem (Rosenberg’s Self-Esteem Scale) believes that there are two types of self-esteem: global self-esteem or how a person feels about themselves overall, and specific self-esteem, how someone appraises a specific part of themselves [56]. The program curriculum targets specific self-esteems related to patient self-advocacy, safer sex communication, medication adherence, and internalized HIV stigma. It is assumed that a combination of work experience and raising specific levels of self-esteem would ultimately raise a person’s global self-esteem. The peer worker training program tools and various support mechanisms helped peers gain new skills, as well as feel better about themselves. Figure 1 displays the predicted paths of psychosocial change according to the program’s conceptual model.

### 2.3. Procedures

From September 2014 through December 2018, nine cohorts of training participants were followed through the end of their six-month internship. This study evaluated survey results from participants living with HIV. To meet the eligibility criteria, individuals had to be medically adherent, stably housed, and in recovery for at least nine months for the peer worker training program and twelve months for internship. Through an application and interview process, staff at the agency decided who would become part of the peer worker training program and who would continue as a paid peer intern. People who did not reach program milestones (completion of the training program and 6 months of internship) were not followed. The study was part of an ongoing program evaluation of an agency’s peer worker training and peer internship program. A secondary dataset was used. The inventories selected for the agency’s peer psychosocial and health survey were grounded in the program’s logic model. A description of the survey inventories is provided below. Study participants took part in the peer psychosocial and health survey at three different time points: Time 1) baseline (first day of peer worker training); Time 2) training completion (final day of training); and Time 3) peer internship (sixth month of their peer internship). Informed consent was provided with each survey. Participants were informed that the survey would be utilized to evaluate the training and peer program. Participants entered and completed the peer worker training program and internship at different time points. Specific numbers for each cohort is listed in Table 1.

### 2.4. Participants

A total of 259 individuals completed a baseline survey. A total of 67 individuals completed a baseline survey only (baseline only group), 137 completed a baseline and training follow-up survey (training group), and 55 individuals completed a baseline, training follow-up, and internship follow-up survey (internship group). Table 1 displays participants by cohort and survey completion.

### 2.5. Measures

The following domains were measured in this study: (1) sociodemographic characteristics (i.e., age, gender, sexual identity, race/ethnicity, education); (2) time since any form of employment; (3) HIV/AIDS status and health indicators (disease stage, years with diagnosis, self-reported HIV viral load and CD4 count, self-assessment of current health, and number of chronic conditions); (4) public benefits received (housing, cash, food, and medical); (5) psychosocial outcomes (depression, HIV internalized stigma, self-esteem); and (6) health-related behavioral outcomes (medication adherence, patient self-advocacy, and safer sex communication apprehension). The psychosocial outcomes measured were informed by the program’s logic model and specific inventories were identified from the literature; inventory descriptions are found below.

***Depression.*** Two inventories were utilized to measure depression during the data collection period, the Beck’s Depression Inventory was utilized from September 2014 through May 2015 (cohorts 1 and 2) and the Patient Health Questionnaire (PHQ-9) from July 2015 through December 2018 (cohorts 3 through 9) [60]. The research team changed depression inventories in July 2015 in order to shorten the length of the peer health and psychological survey. Both measures have been widely used among HIV populations. The PHQ assesses eight DSM-IV diagnoses, and the PHQ-9 encompasses the nine depression questions [60,61]. The PHQ-9 scores range from 0 to 27. Participants are asked how often they experience specific DSM-IV depression symptoms, responses are “not at all”, “several days”, “more than half the days”, and “nearly every day”.

The Beck’s Depression Inventory (BDI) was used with cohorts one and two; it is a 21-item inventory that measures symptoms and attitudes of depression [61]. Individuals rate items from 0 to 3 and scores range from 0 to 66. The inventory was not used during Time 3 because depression was measured by the PHQ-9.

Internalized HIV Stigma. Internalized HIV stigma refers to negative self-perception of self-image because of one’s HIV status [62]. The Internalized Stigma of HIV/AIDS Tool (ISAT) was used to measure internalized HIV stigma [63]. It is a 10-item scale concerning statements related to HIV/AIDS internalized stigma; participants rated the items on a five-point scale, scores range from 5 to 50. 

Self-esteem. Rosenberg’s Self-Esteem Scale is a 10-question measure used widely for people living with HIV [64]. The inventory measures someone’s global self-esteem and uses a four-point Likert scale (1 = strongly disagree, 4 = strongly agree); scores range from 1 to 40. The inventory is widely used among people living with HIV.

HIV Medication Adherence. The AIDS Clinical Trials Group (ACTG) adherence questionnaire is widely used inventory for measuring the adherence of HIV medications [65]. It is a five-item scale assessing: (1) the number of doses missed in the past four days, (2) how closely someone follows specific HIV medication schedules, (3) how closely someone follows specific medication instructions (i.e., must take with food), (4) the last time someone skipped medication, and (5) whether any medications were missed during the last weekend. This study utilized an analysis method described in Reynolds et al., whereby the weekend question is excluded [65]. The researchers created a formula and adherence index normed between 0 and 100, where each question has a different value. For example, missing medication yesterday is a larger part of your score than missing medication four days ago. The inventory’s reliability could not be assessed because each question is not on the same scale.

Safer sex communication apprehension. Safer sex communication apprehension was measured by the safer sex communication apprehension subscale in Elizabeth Babin’s General Sexual Communication Apprehension scale [66]. The subscale contains five statements that are rated on a six-point Likert scale, scores range from 5 to 30. 

Patient Self-Advocacy. The Patient Self-advocacy Scale is a 12-item inventory that measures the involvement of people in their healthcare decision making [67]. The inventory uses a five-point Likert scale, scores range from 5 to 60. The scale is widely used within studies involving PLHIV.

Cronbach’s alpha is used to assess internal consistency for each group at Time 1, Time 2, and Time 3 (Table 2).

### 2.6. Statistical Analysis

Chi-square (*p* < 0.05) and ANOVA tests (*p* < 0.05) were used to determine group differences at baseline between those who completed: (1) the baseline only survey, (2) the baseline and training completion survey, and (3) the baseline, training completion, and internship follow-up surveys. Paired *t*-tests were performed to determine if significant score changes occurred at the individual level between surveys. Independent *t*-tests were used to assess individual inventory changes at training completion among those who participated in six-months of peer internship and those who did not. Without a control comparison group, paired *t*-tests were chosen as the most appropriate statistical methodology to help to evaluate results at the individual level before and after each measured programmatic time point.

## 3. Results

### 3.1. Sociodemographic, Health, and Job History of Program Participants

Table 3 describes the sociodemographic, health, and job histories of program participants at baseline. We reported the complete survey data only as some participants did not complete the entire survey. At baseline, the average age for all participants was 44.6, and 56.37% of participants identified as male, 37.45% female, and 6.18% as transgender. There were 51.8% of participants who identified as heterosexual/straight and 48.19% as LGBTQ. There were 83.14% of participants who had at least a high school education. In total, 94.51% of participants were persons of color, with the majority being Black/African American (61.57%). Chi-squared tests of independence were calculated to compare demographic characteristics by level of program completion. A significant association was not found for gender, age, education level, or sexual identity, but was found for race. Hispanics were more likely to be part of the internship group (30.91%), as compared to the training group (17.65%) and baseline only group (21.57%). Blacks were more likely to be part of the baseline only (60.94%) and training group (68.38%) than the internship group (45.45%). Mixed race/ethnicity participants were more likely to be part of the internship group (18.18%) than the training group (11.03%) or baseline only group (6.25%). White participants were most likely to be in the baseline only group (10.94%) than the training group (2.94%) or internship group (5.45%).

At baseline, the majority of participants’ viral loads were undetectable (83.81%) and they were without an AIDS diagnosis (64.57%). Over half were living with HIV for over 15 years (50.6%) and 19.68% for 5 or less years. Most rated their health as good or better (90.22%) and 37.74% were living with two or more other chronic conditions. Many had a mental health diagnosis (46.12%) and/or a substance abuse history (54.86%). According to the participants, 59.8% had last had a job in the past 5 years. Most were receiving medical benefits (85.27%), housing benefits (71.32%), and cash assistance (78.76%). Chi-squared tests of independence were calculated to compare health characteristics and job histories by level of program completion. Overall, participants in each group were similar demographically, similar in their physical and mental health, and had similar work experiences. A significant association was found for a number of chronic conditions. No other significant differences were observed.

### 3.2. Psychosocial Inventories at Baseline

A one-way between subject ANOVA was conducted to assess if at Time 1 (baseline) there was a statistical difference in psychosocial inventory means between individuals who completed the baseline only survey, the baseline and training completion survey, and those who completed all three. As displayed in Table 4, there was not a significant difference for any of the inventories at the *p* < 0.05 level.

### 3.3. Psychosocial and Health-Related Behavioral Outcome Inventories at Training Completion

A paired-samples *t*-test was conducted to compare each psychosocial inventory by individual at Time 1 (baseline) and at Time 2 (training completion). Table 5 displays paired *t*-test results for the full group. At the *p* < 0.05 level, there was a significant score decrease in depression (PHQ-9) and HIV internalized stigma and a significant score increase in self-esteem, patient advocacy, and medication adherence between Time 1 and Time 2. There was no significant difference for the Beck’s Depression Inventory. Safer sex communication apprehension had no significant score changes.

Table 6 displays the mean inventory scores and the *p*-value for the paired *t*-test sorted by level of program participation. Change in the health-related behavioral outcome inventories at the *p* < 0.05 level differed by those who completed six months of peer internship and those who did not. There was no significant score change in the *health-related behavioral outcome* inventories for the non-internship participants, but there was a significant score increase at the *p* < 0.05 level for patient advocacy and HIV medication adherence and a significant score decrease on the safer sex communication apprehension inventory for those who completed six months of a peer internship. Independent *t*-tests comparing the mean score changes for the training-only participants and the internship and training participants was also conducted. The mean difference for the *health-related behavioral outcome inventories* differed significantly at the *p* < 0.05 level for those who completed six months of peer internship and those who did not.

### 3.4. Psychosocial Inventories for Peer Internship Participants

Internship participants completed surveys at all three time points, Time 1 (baseline), Time 2 (training completion), and Time 3 (6-months of peer internship). Table 7 displays changes in psychosocial inventories between Times 1 and 2; Times 2 and 3; and Times 1 and 3. The behavioral psychosocial inventory for safer sex communication apprehension had a significant score decrease between Times 1 and 2, and Times 1 and 3, but not between Times 2 and 3. Patient self-advocacy had a significant score increase between Times 1 and 2, and 1 and 3, but not between Times 2 and 3. The medication adherence score significantly increased between Times 1 and 2 only. The depression (PHQ-9) and HIV internalized stigma psychosocial inventory scores significantly decreased between Times 1 and 2, and 1 and 3, but not between Times 2 and 3. The self-esteem score significantly increased between Times 1 and 3 only. There was no significant difference for the Beck’s Depression Inventory during any time periods.

## 4. Discussion

The current study evaluated outcomes among participants in an HIV peer worker training and paid internship program. Results demonstrated positive psychosocial and health-related behavioral outcomes after completing the peer worker training program; these changes were maintained throughout the six-month internship program. We discuss each program outcome, its association with employment, and relevant implications for practitioners and researchers.

### 4.1. Psychosocial Outcomes of Peer Worker Training

At the beginning of the study, over 40% of the participants reported at least five years of unemployment and many entered the program struggling with their mental health, trauma, and internalized HIV stigma. This is consistent with past studies indicating that depression and HIV stigma are associated with unemployment among PLHIV [33,34,68,69,70].

**Depression** was significantly decreased in PHQ-9 scores after completing the training program and the internship program. However, there was no significant difference in the BDI scores. Titov et al. compared the psychometrics properties of PHQ-9 and BDI and concluded that BDI captured a greater proportion of individuals with severe depression, compared to PHQ-9 [71]. Therefore, it is possible that the participants in our study had less severe depression and were not captured by BDI. Previous studies also identified an association with employment and decreased depression. Ware found that employment is a protective factor for depression among men living with HIV [72]. Social support has been found as a positive impact on mental health functioning among PLHIV. Participants not only received skills training and support groups during the peer worker training program, but they also formed new peer relationships which increased their social interaction, sense of belonging, and mental health. The quality of employment also has an impact on depression among PLHIV [30]. One study demonstrated that employed PLHIV who had experienced adverse psychosocial work experiences, had similar rates of depression as unemployed individuals [30]. Practitioners may assess individuals’ job security and quality of employment to further understand the impact on their emotional responses. Mental health status is an important HIV prevention and treatment outcome indicator, which has been proven to be connected with health risk behaviors, medication adherence, and retention in care [73]. More research is needed to explore the associations between work and mental health among PLHIV.

**Internalized HIV stigma,** HIV stigma and discrimination are prevalent among PLHIV [74]. Lightner et al. assessed internalized HIV stigma among PLHIV and found that greater internalized stigma is associated with higher levels of perceived employment barriers [75]. Internalized HIV stigma has been found to be associated with poor physical and mental health, lower levels of quality of life, and poor medication adherence. PLHIV who have experienced internalized stigma are less likely to return to work due to fear, trauma, or expectations of future repercussions from others [62]. Our study showed that internalized HIV stigma was significantly decreased after the training program and the internship. In concordance with this finding, previous research has shown that unemployment and underemployment are both associated with disability internalized stigma, and participation in workforce development activities help reduce internalized stigma [76].

Conyers and Boomer examined the role of vocational rehabilitation (VR) access on the goals and objectives of the National HIV/AIDS Strategy and identified that reduced HIV stigma is a key mediator to accessing VR services [38]. Specifically, they found that job confidence had a direct effect on reduced stigma and an indirect effect on use of VR service. HIV-related stigma reduction interventions for PLHIV, community, and employers would be an important strategy to improve employment outcomes for PLHIV [75]. Some programs offered stigma reduction training, but they were not universally utilized and were not specifically designed for the unique needs of PLHIV. It is recommended that job readiness programs should have components that address internalized HIV stigma.

**Self-esteem** was significantly increased in the training-only group. While the peer internship group did not have a significant increase in self-esteem between baseline and training completion, this group did have a significant increase between baseline and internship completion. Escovitz and Donegan evaluated a vocational development program for PLHIV and found that program participation boosted participant’s satisfaction with life [51]. Hergenrather et al. found that participants in work readiness programs for PLHIV had significantly higher self-esteem post-intervention, whereas changes in depression were insignificant [47]. Rosenberg’s model charged that changes in specific self-esteem impact global self-esteem [64,77]. The program’s conceptual model assumed a person’s global self-esteem would be mediated through specific types of self-esteem (i.e., patient self-advocacy, safer sex communication, and medication adherence) during the training period and six-months of hands-on peer work experience. Silván-Ferrero found that a longer length of employment strengthens the relationship between perceived discrimination and self-esteem mediated by internalized stigma among people with mental illness [78]. Further mediation analysis related to self-esteem and internalized stigma with a larger sample size are needed to confirm this theory among peer workers.

### 4.2. Health-Related Behavioral Outcomes of Peer Worker Training

**Medication adherence** was significantly improved during the training program and maintained after six months of a peer internship. Criteria for entering the program include having an undetectable viral load, so the baseline medication adherence score among participants was already high (with an average of 88.9%). Therefore, most of the participants demonstrated a stable health status upon entering the program and, thus, it was anticipated that change would not be significant after the internship.

Previous research also demonstrated a positive impact of employment on medication adherence. In a meta-analysis of medication adherence and employment, Nachega et al. found that employed PLHIV had 27% higher odds of achieving medication adherence, compared to the unemployed PLHIV [79]. Meanwhile, PLHIV who were not working were less likely to engage in HIV treatment and adhere to medications [80]. Research also indicated that PLHIV who had greater medication adherence are more likely to achieve optimal treatment outcomes and higher levels of health-related quality of life [80]. This may be because individuals who are working have a routine schedule as well as a sense of responsibility.

**Self-advocacy** was defined in this study as the involvement of people in their healthcare decision making. Self-advocacy skills are important for PLHIV to gain autonomy and self-control in their treatment plan and health outcomes. Our results showed a significant increase in self-advocacy during the training program and are maintained after six months of a peer internship.

Group interventions have been widely used in facilitating self-advocacy skills. Daniels et al. found that role-play is an effective tool to build self-advocacy skills for PLHIV during their HIV treatment because it provides an opportunity for individuals to share their stories and healthcare experiences [81]. Education and client-centered counseling regarding HIV, HAART, and support services are also essential in building self-advocacy skills [82]. More research is needed on the impact of employment and workforce participation on self-advocacy.

**Safer sex communication apprehension** was significantly increased during the training program and maintained after six months of a peer internship. Safer sex communication related to use of condom, HIV status, sex partner’s HIV risk, and a partner’s reaction to proposing or using condoms. Bond et al. suggested that service providers can facilitate communication strategies that address the cultural norms that influence safer sex practices [83]. In a program outcome study on the effect of peer intervention on partner disclosure, among the participants who agreed to disclose HIV status to their partners, 92.7% of partners who were at risk of HIV transmission were tested, and 96.3% of partners who tested positive were linked to treatment [84]. At baseline, there were no health-related behavioral differences for those whose participation ended at the training and those who became peers. All three behavioral changes happened between baseline and training completion among those who completed both the training program and internship, and there was no significant change for non-internship participants. For people who went on to the internship program, they might have been more ready to work and apply those behavioral changes. Their behaviors could have maintained stable between training program and the internship and that could be the reason why there was no significant change between training completion and internship completion.

Overall, psychosocial changes were mostly associated with participants who continued onto the internship program. There are several possible explanations for this difference. A main purpose of the training was to prepare individuals for a peer internship. HIV service organizations have a long history of utilizing HIV peer workers. Their lived experiences can be advantageous in helping to create behavior change and social support for the individuals they reach through their peer-delivered work [10,12]. Peers in behavioral health settings were expected to “practice what they preach.” Participants who wanted to continue to an internship, may have internalized the behavioral messages of the program more than those who did not want to become an HIV peer worker. Future internship participants may have also been more ready to make the behavioral changes measured in the program. As displayed in Table 3, the psychosocial emotional inventories for internship participants moved in the hypothesized direction during each time point (even though statistical significance was only found for depression and HIV internalized stigma between baseline and training completion, and training completion and internship completion). These findings demonstrate that changes in depression and internalized HIV stigma occur mostly during the training program but are maintained and continue to improve after six months of a peer internship. Some programs measured emotional inventories at baseline, but follow-up measurements were not reported or perhaps not done [44,45]. Hergenrather et al. found their group work readiness participants had significantly higher coping and hope scores between pre- and post-program participation, but changes in anxiety and depression were insignificant [47]. Bedell measured quality of life in their group work readiness intervention and found no significant difference, whereas Escovitz and Donegan found that in their VR plus work placement program, participants’ quality of life had a positive correlation with hours worked [41,49].

On the other hand, behavior change related to HIV self-management was rarely assessed in work readiness literature. A group readiness intervention by Beddell and another by Hergenrather et al. measured the impact of program participation on medication adherence and both found no association [49,51]. Beddell also measured patient self-advocacy and found no effect [49]. Behavior change was probably impacted in this study and not others because participants spent more time in the training and internship programs than the other interventions, and participants were more motivated to make HIV-related behavior changes because of their desire to become peers.

### 4.3. Limitations

Data collection for this study was part of a program evaluation and was not designed to be a quasi-experimental study. Selection bias is a major threat to the internal validity of the study. The studied agency chose who would become part of the training program and become a paid peer intern. There might be staff bias during the application and interview process. Another potential source of selection bias is the participants themselves. Those who derived the most benefit from the program might have been more likely to complete the training program and to continue on to completing the internship. In addition, people who did not complete program milestones (completion of training program and six months of peer internship) were not part of the program evaluation. Although statistically, all three baseline groups (baseline only, training completers, and internship participants) were similar demographically and had similar baseline psychosocial inventory scores. This study design limitation made it difficult to decipher if participating in the internship program truly provided additional benefits due to lack of appropriate control groups. Without being able to control for why individual participants were chosen to be part of the two programs, it is hard to decipher true program impact. In addition, the sample size was rather small, and participants were only those residing in New York City. Therefore, it might not be necessarily generalizable to other groups of PLHIV. Data collection took place over many years; programmatic changes occurred cycle to cycle, partly contributing to the fluctuating number of valid observations.

Changes included peer pay and changes in requirements of skill sets needed for positions. Furthermore, data collection was self-report. Most participants applied to the training program in the hope of finding employment, and most peer interns hope to retain their peer employment for more than one cycle; it is possible that survey responses were biased due to their employment desires at the agency.

## 5. Conclusions

People living with HIV are disproportionately impacted by structural social determinants of health, such as racial and ethnic identities, gender and sexual identities, HIV stigma, poverty, substance use, domestic violence, homelessness, incarceration, and unemployment, each contributing to adverse impacts on their physical and psychosocial well-being [85,86,87]. The participants in the study represented the historically marginalized groups in the US based on their race, ethnicity, and gender and sexual identities. Systemic health disparities are globally recognized, with increased rates of mortality and disease progression disproportionately falling upon marginalized populations, primarily low-income communities of color. Stigma toward substance use, mental illness, and HIV, racism, and homophobia are all structural barriers to employment. HIV peer worker training is a strategy to combat these inequities by providing toolkits to employment and enhancing self-esteem and confidence. In addition, the involvement in the HIV workforce provides PLHIV opportunities for future paid employment. Employment and workforce participation are associated with HIV prevention and treatment efforts. Our study identified positive psychosocial and health-related behavioral impacts associated with participating in an HIV peer worker training and peer internship program. Individuals not only developed work skills but also self-management skills. Although the study design has limitations, study findings warrant further exploration of the peer worker training and internship program. To better understand long-term outcomes, such as paid employment and quality of life of the peers, future studies should use a longitudinal approach and include a comparison group.

This study assesses the impact of the foundational peer worker training course and paid internship requirements of the state certification on an individual’s psychosocial outcomes. Due to the possible effects on public benefits and an ability to maintain medical care, returning to work full-time might not be the recommended course of action for everyone living with HIV, and therefore was not the goal for program participants. Individuals’ perception about themselves might have an impact on their HIV treatment adherence and overall quality of life; therefore, the psychosocial impacts of the peer worker training and internship programs would be important indicators to evaluating a work readiness peer program. Topping conducted a meta-analysis of peer intervention and education programs and suggested that program structure, funding, management, supervision, and retention are all related to program outcomes [88]. Future studies may focus on structural factors of HIV service organizations. HIV service organizations and programs may provide work readiness skills and job opportunities for PLHIV, while concurrently improving emotional and behavioral psychosocial outcomes, which are critical investments for PLHIV. Structurally based interventions for PLHIV, such as housing and food assistance, have been federally funded for decades due to their impact on social determinants of health [82,88]. AIDS activists were able to achieve this funding due to the plethora of research positively supporting their impact. Work readiness programs are a viable structural prevention technique for PLHIV because work improves the quality of life for PLHIV.

Investing in work readiness could save the government millions of dollars through helping PLHIV to become independent from benefits. There has been a dearth of the literature measuring the impact of work readiness programs for PLHIV in the US; fewer than a dozen evaluated programs have been published since the introduction of highly active antiretroviral therapy (HAART) [42,43,44,45,46,47,48,49,50,51,54,55,56]. In New York State, the Office of Temporary Disability Assistance (OTDA) authorized a law called “peer income disregard.” This law allows HASA recipients to collect income as a paid peer intern at an AIDS service organization in addition to their benefits with no time limitations [89]. For example, a peer can earn their peer wage and continue to receive housing subsidies, SNAP benefits, and Medicaid. Individuals who choose other types of employment lose parts of their benefits. Individuals receiving SSDI are eligible for Ticket to Work, a time limited program for social security beneficiaries to receive pay and benefits while they transition to work. SSI recipients are not eligible for any of these programs. The OTDA peer income disregard policy in New York allows for peers to have their basic needs met from HASA and the income from peer work increases their financial security [90]. In addition, expanding the peer income disregard to other states, expanding the time for Ticket to Work for SSDI recipients, and creating a similar income disregard allowance for people with SSI is strongly recommended. Research is needed to explore the relationships between the use of these benefits program and employment outcomes.

## Figures and Tables

**Figure 1 ijerph-20-04322-f001:**
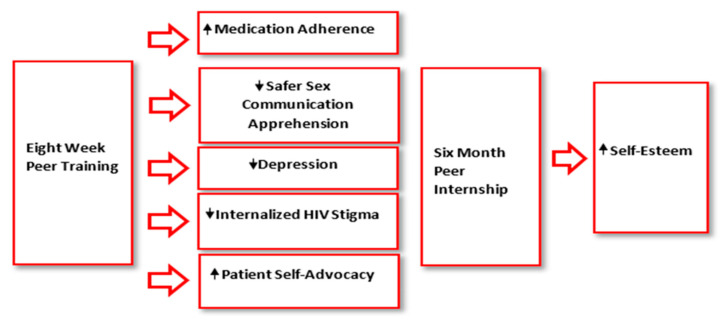
Paths of psychosocial change.

**Table 1 ijerph-20-04322-t001:** Participants by cohort and survey completion (N = 259).

	Number of Participants Who Completed Surveys during Program Milestones					
	Baseline Only Participants		Training Completion Participants		Training and Internship Participants	
	(N = 67)		(N = 137)		(N = 55)	
	Survey Timeframe	*n*	Survey Timeframe	*n*	Survey Timeframe	*n*
Cohort One	September 2014	13	November 2014	9	June 2015	9
Cohort Two	March 2015	6	May 2015	8	December 2015	9
Cohort Three	September 2015	8	November 2015	12	June 2016	11
Cohort Four	March 2016	6	May 2016	20	December 2016	5
Cohort Five	September 2016	7	November 2016	16	June 2017	4
Cohort Six	March 2017	7	May 2017	18	December 2017	4
Cohort Seven	September 2017	6	November 2017	21	June 2018	4
Cohort Eight	March 2018	7	May 2018	15	December 2018	9
Cohort Nine	September 2018	7	November 2018	18	-	-
	Total	67		137		55

**Table 2 ijerph-20-04322-t002:** Internal consistency for each measure.

Measures	Cronbach’s α Time 1	Cronbach’s α Time 2	Cronbach’s α Time 3
Patient Health Questionnaire (PHQ-9)	0.86	0.89	0.85
Beck’s Depression Inventory (BDI)	0.93	0.96	NA
The Internalized Stigma of HIV/AIDS Tool (ISAT)	0.88	0.88	0.89
Rosenberg’s Self-Esteem Scale	0.83	0.81	0.84
General Sexual Communication Apprehension scale	0.73	0.68	0.67
The Patient Self-advocacy Scale	0.76	0.70	0.68

**Table 3 ijerph-20-04322-t003:** Baseline sociodemographic characteristics of participants.

Baseline Participants					
(N = 259)					
Category	n	%	Category	n	%
Baseline Demographics					
Gender			Age		
Male	146	56.37	18–34	54	21.26
Female	97	37.45	35–44	65	25.59
Transgender	16	6.18	45–54	83	32.68
Education Level			55–64	52	20.47
<High School	43	16.86	Race/Ethnicity		
HS Graduate or GED	170	66.67	Black	157	61.57
College	42	16.47	Hispanic	55	21.57
Sexual Identity			White	14	5.49
Straight	129	51.8	Mixed	29	11.37
LGBTQ	120	48.19			
Baseline health characteristics					
AIDS Diagnosis			Viral Load		
Yes	90	35.43	Undetectable	207	83.81
No	164	64.57	Detectable	40	16.19
Years Living with HIV			Self-Rated General Health in the Past Year		
0–5 Years	49	19.68	Excellent	46	20.44
6–10 Years	32	12.85	Very Good	90	40
11–15 Years	42	16.87	Good	67	29.78
16–25 Years	82	32.93	Fair or Poor	22	9.78
> 25 Years	44	17.67	Alcohol and/or Drug Addiction History		
Number of Chronic Conditions Other than HIV			Yes	141	54.86
None	81	31.52	No	116	45.14
One	79	30.74	Mental Health Diagnosis		
More than Two	97	37.74	Yes	119	46.12
			No	139	45.14
Baseline work history and benefits received					
Years since last part-time or full-time employment			Public Benefits Received		
Past 2 Years	99	41.08	Medical *	220	85.27
Past 5 Years	45	18.67	Housing **	184	71.32
Past 5–10 Years	43	17.84	Cash ***	204	78.76
Over 10 Years	35	14.52	SSI or SSD	30	14.02
Never	19	7.88			

***** Medicaid, Medicare. ** NYC Housing for PLWHA or Public Housing. *** Food Stamps, Cash assistance for PLWHA.

**Table 4 ijerph-20-04322-t004:** Psychosocial inventories at baseline by level of program completion.

	Participants by Program Completion Type							
	Baseline Only Participants		Training Completion Participants		Training and Internship Participants		ANOVA	
	Mean (SD)	Valid *n*	Mean (SD)	Valid *n*	Mean (SD)	Valid *n*	f-Value	*p*-Value
Safer Sex Com. Apprehension	12.59 (6.55)	60	11.83 (5.74)	131	12.06 (5.57)	54	0.34	0.71
Patient Self-Advocacy	41.25 (8.25)	57	43.3 (5.84)	118	42.07 (6.79)	50	1.93	0.15
HIV Medication Adh.	92.34 (11.98)	54	88.2 (18.6)	110	92.34 (11.98)	48	1.37	0.26
Beck’s Depression Inv.	14.74 (12.20)	17	12.95 (11.64)	17	11.75 (9.06)	18	0.32	0.72
PHQ-9 Depression Inv.	5.28 (5.72)	46	4.68 (4.72)	118	5.38 (4.85)	37	0.43	0.65
HIV Intern. Stigma	24.59 (9.92)	65	24.02 (8.49)	134	26.93 (9.10)	55	2.05	0.13
Self-Esteem	21.92 (5.35)	65	21.49 (5.0)	136	20.81 (4.62)	55	0.73	0.48

**Table 5 ijerph-20-04322-t005:** Change in psychosocial inventories for individuals who completed the PREP training, paired *t*-test.

	Survey Period			
	Time 1	Time 2	Paired *t*-Test	
Psychosocial Inventory (Valid N)	Mean (SD)	Mean (SD)	t-value	*p*-value
Safer Sex Communication Apprehension (176)	11.95 (5.72)	11.18 (5.54)	−1.66	0.1
Patient Self-Advocacy (153)	42.84 (8.25)	43.94 (6.79)	2.08	0.04
HIV Medication Adherence (149)	88.01 (11.98)	90.88 (11.98)	2.2	0.03
Beck’s Depression Inventory (35)	11.75 (10.25)	10.80 (11.58)	−0.9	0.38
PHQ-9 Depression Inventory (150)	4.82 (4.70)	3.04 (4.19)	−4.49	<0.001
HIV Internalized Stigma (185)	25.06 (8.74)	23.02 (8.79)	−3.79	<0.001
Self-Esteem (186)	21.33 (4.87)	22.49 (4.88)	4.03	<0.001

**Table 6 ijerph-20-04322-t006:** Change in psychosocial inventories scores from baseline to training completion, by level of program participation.

	Participants by Program Completion Type							
	Training Completion Only Participants				Training and Internship Participants			
		Time 1	Time 2	Paired *t-*Test		Time 1	Time 2	Paired *t-*Test
Inventory	Valid *n*	Mean (SD)	Mean (SD)	*p*-value	Valid *n*	Mean (SD)	Mean (SD)	*p*-value
Safer Sex Com. Apprehen.	123	11.95 (5.8)	11.84 (5.69)	0.83	52	12.2 (5.53)	9.66 (5.74)	0.001
Patient Self-Advoc.	108	43.07 (6.0)	43.47 (6.74)	0.5	48	41.95 (6.8)	44.65 (5.19)	2
HIV Med. Adh.	103	88.44 (15.8)	89.51 (13.92)	0.77	53	88.81 (17.0)	93.94 (13.98)	0.02
Beck’s Depression	17	12.94 (12.99)	12.73 (11.64)	0.94	18	11.75 (9.06)	8.86 (10.00)	0.26
PHQ-9 Depression	86	4.86 (5.10)	3.00 (3.82)	0.001	35	5.21 (4.9)	2.77 (3.42)	<0.001
HIV Intern. Stigma	131	24.23 (8.46)	22.45 (8.82)	0.001	53	27.15 (9.05)	24.67 (8.48)	0.003
Self-Esteem	132	21.36 (4.98)	22.70 (4.93)	0.001	53	20.7 (4.58)	21.85 (4.69)	0.07

**Table 7 ijerph-20-04322-t007:** Change in psychosocial inventories for internship participants.

	Survey Period			Paired *t*-Test		
	Baseline	Training Completion	Internship	Time 1–2	Time 2–3	Time 1–3
Psychosocial Inventory (Valid N)	Mean (SD)	Mean (SD)	Mean (SD)	*p*-value	*p*-value	*p*-value
Safer Sex Communication Apprehension (N = 52)	12.2 (5.53)	9.66 (5.74)	10.22 (5.03)	0.001	0.21	0.02
Patient Self-Advocacy (N = 48)	41.95 (6.8)	44.65 (5.19)	45.17 (4.82)	0.02	0.58	<0.001
HIV Medication Adherence (N = 53)	88.81 (17.0)	93.94 (13.98)	92.83 (12.87)	0.02	0.33	0.37
Beck’s Depression Inventory (N = 18)	11.75 (9.06)	8.86 (10.00)		0.26		
PHQ-9 Depression Inventory (N = 35)	5.21 (4.9)	2.77 (3.42)	2.63 (3.03)	<0.001	0.78	0.003
HIV Internalized Stigma (N = 53)	27.15 (9.05)	24.67 (8.48)	23.06 (8.02)	0.003	0.18	<0.001
Self-Esteem (N = 53)	20.7 (4.58)	21.85 (4.69)	22.74 (4.64)	0.07	0.23	0.01

## Data Availability

Not applicable.

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
