# Peer review of "Psychosocial and Health-Related Behavioral Outcomes of a Work Readiness HIV Peer Worker Training Program"

_ijerph, 2023, doi:10.3390/ijerph20054322_

Round 1
Reviewer 1 Report
Please see attached

Author Response
Response to Reviewer 1 Comments
Point 1: P. 3 line 97 consider saying a bit more about the stigma that may be associated with utilizing services, including VR among PLWHIV. This is a critical barrier that remains understated.
Response 1: We added “HIV-related stigma” as one of the factors of under utilization of VR services in the section of “Vocational development and intervention for PLHIV”.
Point 2: There is very little mentioned about the barrier of alcohol and substance use despite the fact that over 54% of participants report a history of alcohol or drug addiction. For this reviewer, the findings regarding the potential for the combination training is even more important for this reason. Please say more.
Response 2: We added “substance use” as one of the barriers to employment in the “employment as a social determinant of health” section.
Point 3: Table 2 title says “Baseline … by level of program completion” but it is total only. Offering a Table truly by program completion would be far more helpful to the reader.
Response 3: We fixed the title of Table 3 (previous Table 2).
Point 4: p. 4, line 176, consider saying something about the fact that you were interested in examining the additive effects of the peer training plus peer internship combination.
Response 4: We added the description about our intention to examine the additive effects of the peer training plus peer internship combination.
Point 5: Directly under your Methods heading consider a 1-2 sentence summary of your design, e.g. “This a secondary data analysis of an evaluation conducted between xx and xx, by XXX to examine the psychosocial and health related behavioral outcomes among XX individuals participating in a XXX program.
Response 5: We added a paragraph summarizing the study design based on one of the reviewer’s recommendations.
Point 6: Have there been any other evaluations of this NYC Peer Based Training? Is this the first? It would be helpful to situate it within what is known regarding these efforts and clarify this for the reader in the Background section. IF this is the first, then important to say so, too!
Response 6: Yes it is the first evaluation and we added a description in the first paragraph of methods section.
Point 7: Authors should mention the assignment by staff to which group here. It isn’t mentioned until the Limitations but belongs here in the Methods section, so the reader understands this up front. 2 RE: the requirements to gain entry into the program, 9 months for training and 12 months for internship, may also have had a significant impact on who is in which program, and the outcomes, correct? This is important to address in your limitations and mention in Discussion.
Response 7: We added the participants' selection process in the “procedure” section, which is consistent with the selection bias we discussed in the “limitation” section.
Point 8: Line 242, no need to repeat sentence “Nine cohorts…” you said it already. Is there any information available to understand attrition between these three groups? It would be very helpful to say something about this – or in the absence of this information, perhaps in the Discussion, talk about what the programmatic folks believe happened here.
Response 8: We revised Table 1 to include the nine cohorts of participants to clarify one of the questions about the participants.
Point 9: What do stars on Table 2 mean? Footnote needed. Tables provide Ns for each variable and category, but they differ from the total, indicating missingness.
Response 9: we added footnote in Table 3 (previously Table 2).
Point 10: Mention the fact that authors are using complete data only and that there is data missing in your Limitations section. Not at all unusual for an evaluation study, but still helpful to note.
Response 10: We added a description in the first paragraph of the results section.
Point 11: This section could be *much* leaner and to the point, which would make it more impactful. Avoid restating your results/findings as this was already done in the Results section. Instead, tell us what the findings mean in relation to other studies. How is each consistent, or not, with related studies. This section needs the most editorial attention. Consider more summary paragraphs and not necessarily going variable by variable. Or if the outline stays variable to variable, shorten by taking out any restatement of measures or findings.
Response 11: We reduced and shortened the description of our findings in the discussion section and focused on the related studies in the field and the implications for practice and future research.
Point 12: Given the fact that this is an evaluation study without randomization or control group, please soften ALL language that suggests any causal relationship. It is impossible* to know if the intervention program was responsible for the findings, but this is an important study and findings, and does not need to be oversold. The trends and findings speak well to the potential value of the intervention and calls for more formal assessment of the program to implement with policy supports. *For example, lines 604-605: “Our study identified positive psychosocial and health-related behavioral impacts of participating in a HIV peer training…” Should say “associated with” and not “of” .
Response 12: We changed all the languages related to causal relationships across the manuscript.
Point 13: Finally, the authors avoid stating more clearly that we are talking about a majority of Black and Brown individuals at a time when anti-Black racism is emergent in our nation, City, and in our social policies. If they are comfortable making stronger claims about the importance of this research and the policy implications for health equity, it would be most welcomed.
Response 13: We added a paragraph discussing racial disparities and equity issues in the conclusion section.
Reviewer 2 Report
REVIEW REPORT FOR THE STUDY “PSYCHOSOCIAL AND HEALTH-RELATED BEHAVIORAL OUTCOMES OF A WORK READINESS HIV PEER TRAINING PROGRAM”
Journal: Int. J. Environ. Res. Public Health
The paper "Psychosocial and Health-Related Behavioral Outcomes of a Work Readiness HIV Peer Training Program", performs a study on a longitudinal evaluation of the peer training and internship program. Emotional psychosocial outcomes such as depression or HIV internalized stigma and behavioral psychosocial outcomes like medication adherence or patient self-advocacy were measured in this study. Paired t-tests and ANOVA analysis were performed to determine if significant score changes occurred at the individual level between surveys. This paper is the result of work that has constituted the thesis of author Erin R. McKinney-Prupis at the CUNY School of Public Health titled “Psychosocial impacts and employment preparedness of participating in an HIV peer program: Implications for tailoring a vocational counseling model for PLWH”.
Title and summary. The title and abstract express well the object of study and results of the article.
Structure of the article. The contents are well organized and they adhere to the IMRaD structure. It includes a theoretical framework of the research problem but at this point, I suggest the authors incorporate some other bibliographic references that I miss in the text:
Remien RH, Patel V, Chibanda D, Abas MA. Integrating mental health into HIV prevention and care: a call to action. J Int AIDS Soc. 2021 Jun;24 Suppl 2(Suppl 2):e25748. doi: 10.1002/jia2.25748. PMID: 34164925; PMCID: PMC8222846.
Lambert SM, Debattista J, Bodiroza A, Martin J, Staunton S, Walker R. Effective peer education in HIV: defining factors that maximise success. Sex Health. 2013 Aug;10(4):325-31. doi: 10.1071/SH12195. PMID: 23725575..
Han H-R, Kim K, Murphy J, Cudjoe J, Wilson P, Sharps P, et al. (2018) Community health worker interventions to promote psychosocial outcomes among people living with HIVÐA systematic review. PLoS ONE 13(4):e0194928. https://doi.org/10.1371/journal. pone.0194928.
Focusing on the opportunity of the study, it must be said that it is useful work since it promotes psychosocial outcomes among people living with HIV.
Materials and methods.
Regarding the material and methods section, the methodology is tailored to the object of study and the objectives while it has been validly applied to guarantee the results.
However, I would like to suggest to the authors, with the intention of reinforcing the choice of methodology, to explain the choice of evaluation methodologies and why they have not chosen logistic regression or interrupted time series regression which could be applicable to the evaluation of an intervention over time.
Results.
The results are significant and they are presented in an adequate and understandable way not only through narration but also with self-explained tables and figure that are also well elaborated in terms of presentation. The results justify and relate to the objectives and methods and the results are of sufficient interest.
Discussion.
The discussion appropriately compares the study results with other works, highlighting the main study findings.
However, I would propose the inclusion of three bibliographic references in the discussion section:
Topping, K.J. Peer Education and Peer Counselling for Health andWell-Being: A Review of Reviews. Int. J. Environ. Res. Public Health 2022, 19, 6064. https://doi.org/ 10.3390/ijerph19106064.
Dawson-Rose C, Gutin SA, Mudender F, Hunguana E, Kevany S (2020) Effects of a peer educator program for HIV status disclosure and health system strengthening: Findings from a clinic-based disclosure support program in Mozambique. PLoS ONE 15(5): e0232347. https://doi.org/10.1371/journal.pone.0232347
Wu, S., Roychowdhury, I. & Khan, M. Evaluations of training programs to improve human resource capacity for HIV, malaria, and TB control: a systematic scoping review of methods applied and outcomes assessed. Trop Med Health 45, 16 (2017). https://doi.org/10.1186/s41182-017-0056-7.
Bibliography.
The 14.28% of the bibliography cited in the study belongs to the previous five years.
Overall, it is an interesting study and should be considered for publication in Int. J. Environ. Res. Public Health, once the minor revisions proposed have been resolved.
Author Response
Thank you for providing detailed feedback for our manuscript, entitled “Psychosocial and Health-Related Behavioral Outcomes of a Work Readiness HIV Peer Worker Training Program.”We had carefully reviewed your feedback and revised our manuscript. Please find the revised manuscript in the system. We are also providing a summary responding to your comments in this response.
Point 1: It includes a theoretical framework of the research problem but at this point, I suggest the authors incorporate some other bibliographic references that I miss in the text.
Response 1: we included these bibliographic references in the introduction section.
Point 2: I would like to suggest to the authors, with the intention of reinforcing the choice of methodology, to explain the choice of evaluation methodologies and why they have not chosen logistic regression or interrupted time series regression which could be applicable to the evaluation of an intervention over time.
Response 2: The reason why we chose a program evaluation methodology using paired t-tests was that we did not have a comparison group and therefore the most valuable information to assess was change by person, which paired t-test does well. We did not use regression or time series methodologies as the numbers of participants in each group are different.
Point 3: I would propose the inclusion of three bibliographic references in the discussion section.
Response 3: We added these bibliographic references in the discussion section.
Thank you again for reviewing our manuscript. We hope to hear back from you soon.
Reviewer 3 Report
Thank you for the opportunity to review your submission. The manuscript was very well written and a pleasure to read. As discussed by the authors, selection bias is a major threat to the validity of the study, as is the lack of a control group. However, the data themselves are interesting and support the authors' interpretation that a work-readiness peer training program for PLHIV. I have a few minor comments/suggestions:
1. line 190 - STI's should be STIs.
2. The introduction contains a description of the compositions of other peer training programs, which does not appear overly relevant to the study findings or discussion. I suggest removing/reducing this section unless the the discussion is modified to include an analysis of what elements of the training program the authors suppose are likely to have had the greatest impact.
3. The data presented do not support the findings that the internship program resulted in further improvement in levels of depression, internalized HIV stigma, medication adherence, self advocacy, safer sex communication apprehension. At most, the internship program helped maintain gains made during the training program, although this is difficult to determine as the training only group were not surveyed at time 3 (i.e. there was no control). Suggest modifying the text around the differences between measures at time 1 and 3 to remove the implication of continued improvement/gains after the internship program. (approx lines 431, 455, 492, 513, 523)
4. In the limitations section, it would be good to clarify that it is difficult to determine if the internship program provided any benefit to participants (aside from self esteem) in either maintaining of building upon gains made by participants during the training program due to a lack of appropriate control groups.
5. Also in the limitations section, the authors rightly point out that the lack of information about how or why participants were selected for the internship program may introduce a degree of selection bias. Another potential source of selection bias is the participants themselves. Those who derived the most benefit from the program may have been more likely to complete the training program and continue on to completing the internship.
Author Response
Thank you for providing detailed feedback for our manuscript, entitled “Psychosocial and Health-Related Behavioral Outcomes of a Work Readiness HIV Peer Worker Training Program.”We had carefully reviewed your feedback and revised our manuscript. Please find the revised manuscript in the system. We are also providing a summary response here:
Point 1: Line 190 - STI's should be STIs.
Response 1: Revised.
Point 2: The introduction contains a description of the compositions of other peer training programs, which does not appear overly relevant to the study findings or discussion. I suggest removing/reducing this section unless the discussion is modified to include an analysis of what elements of the training program the authors suppose are likely to have had the greatest impact.
Response: We reduced the content related to other peer training and vocational training programs; however, the current literature does not include any studies on peer worker training. Therefore, we cited some literature from the field of vocational rehabilitation as it is relevant to our work readiness peer training program.
Point 3: The data presented do not support the findings that the internship program resulted in further improvement in levels of depression, internalized HIV stigma, medication adherence, self advocacy, safer sex communication apprehension. At most, the internship program helped maintain gains made during the training program, although this is difficult to determine as the training only group were not surveyed at time 3 (i.e. there was no control). Suggest modifying the text around the differences between measures at time 1 and 3 to remove the implication of continued improvement/gains after the internship program. (approx lines 431, 455, 492, 513, 523)
Response 3: We revised the text around the differences between measures at time 1 and 3 to discuss the significant change between baseline and peer training program that was maintained through the end of the 6-month internship.
Point 4: In the limitations section, it would be good to clarify that it is difficult to determine if the internship program provided any benefit to participants (aside from self esteem) in either maintaining of building upon gains made by participants during the training program due to a lack of appropriate control groups.
Response 4: We added a limitation of lack of control groups in the limitation section.
Point 5: Also in the limitations section, the authors rightly point out that the lack of information about how or why participants were selected for the internship program may introduce a degree of selection bias. Another potential source of selection bias is the participants themselves. Those who derived the most benefit from the program may have been more likely to complete the training program and continue on to completing the internship.
Response 5: We added a limitation of selection bias based on participants’ self-motivation.
Thank you again for reviewing our manuscript. We hope to hear back from you soon.